# The Relationship between Dietary Vitamin K and Depressive Symptoms in Late Adulthood: A Cross-Sectional Analysis from a Large Cohort Study

**DOI:** 10.3390/nu11040787

**Published:** 2019-04-05

**Authors:** Francesco Bolzetta, Nicola Veronese, Brendon Stubbs, Marianna Noale, Alberto Vaona, Jacopo Demurtas, Stefano Celotto, Chiara Cacco, Alberto Cester, Maria Gabriella Caruso, Rosa Reddavide, Maria Notarnicola, Stefania Maggi, Ai Koyanagi, Michele Fornaro, Joseph Firth, Lee Smith, Marco Solmi

**Affiliations:** 1Medical Department, Geriatric Unit, Azienda ULSS (Unità Locale Socio Sanitaria) 3 “Serenissima”, 30031 Dolo-Mirano District, Italy; francesco.bolzetta@gmail.com (F.B.); alberto.cester@aulss3.veneto.it (A.C.); 2Aging Branch, Neuroscience Institute, National Research Council, 35128 Padua, Italy; marianna.noale@in.cnr.it (M.N.); stefania.maggi@in.cnr.it (S.M.); 3National Institute of Gastroenterlogy, Research Hospital, IRCCS De Bellis, Castellana Grotte, 70013 Bari, Italy; gabriella.caruso@irccsdebellis.it (M.G.C.); rosa.reddavide@irccsdebellis.it (R.R.); maria.notarnicola@irccsdebellis.it (M.N.); 4South London and Maudsley NHS Foundation Trust, Denmark Hill, London SE5 8AZ, UK; brendon.stubbs@kcl.ac.uk; 5Faculty of Health, Social care and Education, Anglia Ruskin University, Bishop Hall Lane, Chelmsford CM1 1SQ, UK; 6Institute of Psychiatry, Psychology and Neuroscience (IoPPN) King’s College London, De Crespigny Park, London SE5 8AF, UK; 7Primary Care Department, Azienda ULSS20 Verona, 37122 Verona, Italy; aisamaisa@gmail.com; 8Primary Care Department, Azienda USL Toscana Sud Est, 58100 Grosseto, Italy; eritrox7@gmail.com; 9Primary Care Department, Aziendale AAS3 Alto Friuli – Collinare – Medio Friuli, 33013 Udine, Italy; celottostefano@gmail.com; 10University of Siena, 53100 Siena, Italy; chiaracacco@gmail.com; 11Research and Development Unit, Parc Sanitari Sant Joan de Déu, Fundació Sant Joan de Déu, CIBERSAM, 28029 Barcelona, Spain; koyanagi1117@hotmail.com; 12New York State Psychiatric Institute, Columbia University, New York, NY 10027, USA; dott.fornaro@gmail.com; 13NICM Health Research Institute, University of Western Sydney, Penrith, NSW 2751, Australia; joefirth@gmail.com; 14Division of Psychology and Mental Health, University of Manchester, Manchester M13 9PL, UK; 15The Cambridge Centre for Sport and Exercise Sciences, Department of Life Sciences, Anglia Ruskin University, Cambridge CB1 1PT, UK; Lee.Smith@anglia.ac.uk; 16Department of Neuroscience, University of Padova, 35122 Padova, Italy; marco.solmi83@gmail.com; 17Padova Neuroscience Center, University of Padova, 35122 Padova, Italy

**Keywords:** vitamin K, depression, Osteoarthritis Initiative, mental health, diet, nutrition

## Abstract

Few studies assessed the associations between dietary vitamin K and depressive symptoms. We aimed to investigate the association between dietary vitamin K and depressive symptoms in a large cohort of North American People. In this cross-sectional analysis, 4,375 participants that were aged 45–79 years from the Osteoarthritis Initiative were included. Dietary vitamin K intake was collected through a semi-quantitative food frequency questionnaire and categorized in quartiles. Depressive symptoms were diagnosed using the 20-item Center for Epidemiologic Studies-Depression (CES-D) ≥ 16. To investigate the associations between vitamin K intake and depressive symptoms, logistic regression analysis were run, which adjusted for potential confounders. Overall, 437 (=10%) subjects had depressive symptoms. After adjusting for 11 confounders, people with the highest dietary vitamin K intake had lower odds of having depressive symptoms (OR = 0.58; 95%CI: 0.43–0.80). This effect was only present in people not taking vitamin D supplementation. In conclusion, higher dietary vitamin K intake was significantly associated with a lower presence of depressive symptoms, also after accounting for potential confounders. Future longitudinal research is required to explore the directionality of the association.

## 1. Introduction

The prevalence of depressive disorders among old people in the United States is approximately 4.5% [1]. Late life depression significantly decreases the quality of life [2] and it is often accompanied by significant medical burden and disability [2]. Moreover, it is related to significantly increased risk of stroke [3], myocardial infarction [4], and finally mortality [3]. Although major depression is a very common disease, the pathogenesis is still presently unknown. Depression in the elderly is complex and it may occur for various reasons and under different conditions. It has already been pointed out that diet and nutrition, particularly vitamins, may influence depression [5].

Being especially known for its role in blood coagulation, there is now evidence that vitamin K has important actions in the nervous system. In particular, it has been implicated in the regulation of sphingolipid metabolism and in protection against oxidative stress in the brain [6]. Notably, it has been demonstrated that vitamin K concentrations in the brain are influenced by diet in a way that reflects its intake, even if these findings are limited to a few animal models [7]. On the other hand, concerns have been expressed that the current recommended intake of vitamin K (120 ug/d for men and 90 ug/d for women) might not provide sufficient amounts for fulfilling all the functions of vitamin K proteins beyond those that are involved in coagulation [8].

In a previous study, vitamin K deficiency, as induced by dietary depletion or by warfarin treatment, was associated with hypoactivity and a lack of exploratory behavior in rats [9]. Moreover, in another study, vitamin K supplementation antagonized depression-like behavior in rats with a dietary model of metabolic syndrome [10]. Currently, there are only two studies on human beings, with contrasting results to the best of our knowledge. Recently, a Japanese team examined the relationship between vitamin K intake and depressive symptoms in elderly people, finding a positive relation between low vitamin K intake and depression [5]. On the contrary, a study in Spanish children found no such association [11].

Given this background, the aim of the present study was to investigate the association between vitamin K intake and the presence of depressive symptoms in a large cohort of older adults that reside in North America who are at risk of or who have osteoarthritis. Based on previous studies, we hypothesized that persons with higher intake of vitamin K would be at a lower risk of presenting depressive symptoms.

## 2. Materials and Methods

### 2.1. Data Source and Subjects

The Osteoarthritis Initiative database [12] is available for public access at http://www.oai.ucsf.edu/. The specific datasets that were utilized were registered during the baseline and screening evaluations (V00). The OAI (Osteoarthritis Initiative) includes people at high risk of knee osteoarthritis or having knee osteoarthritis, who were recruited at four clinical centers in the USA (Baltimore, MD, USA; Pittsburgh, PA, USA; Pawtucket, RI, USA; and, Columbus, OH, USA) between February 2004 and May 2006. People were eligible if they: i) had knee osteoarthritis and reported knee pain in a 30-day period in the past 12 months or ii) were at high risk of developing knee OA (e.g., overweight/obese, knee injury/operation, parents/siblings with total knee replacement, frequent knee-bending activities that increase risk, and hand/hip osteoarthritis) [12]. 

All of the participants provided written informed consent. The Osteoarthritis Initiative study protocol was approved by the institutional review board of the Osteoarthritis Initiative Coordinating Center, University of California at San Francisco.

### 2.2. Dietary Vitamin K Intake (Exposure)

The semi-quantitative Block Brief 2000 food frequency questionnaire (FFQ) at baseline, which is a validated and widely used method [13,14], was used for collecting information regarding dietary habits over the previous year. This validated tool contains a food list of 70 items and it was designed to assess the individual’s food and beverage consumption over the past year, including the consumption of supplements. Frequency of food consumption of the included items was reported at nine levels (“never” to “every day”). The vitamin K intake was then calculated as the sum of vitamin K supplementation (if any) and of dietary intake of vitamin K, as assessed by the diet recall questionnaire, the Block Brief 2000 FFQ. NutritionQuest (http://www.nutritionquest.com/assessment/), using standard values for amounts of vitamin K across an array of foods automatically calculated the Vitamin K scores for this study. The foods with particularly high reference levels of vitamin K are leafy vegetables, such as kale, broccoli, and spinach, with lower (but still notable) levels in fruits and berries and fish. 

Dietary vitamin K intake was arbitrarily categorized in four quartiles while using the following cut-offs: 83, 138, 232 ug/day. Moreover, we modelled the dietary vitamin K intake as per 100 ug/day increment. 

### 2.3. Outcome (Depressive Symptoms)

The presence of depressive symptoms was assessed from the 20-item Center for Epidemiologic Studies-Depression (CES-D) instrument [15]. The range of possible values for this scores is 0 to 60, where higher scores indicate more depressive symptoms [15]. A cut-off of 16 was used for the presence of depressive symptoms [16].

### 2.4. Covariates

We identified numerous potential confounders that may influence the relationship between dietary vitamin K and depressive symptoms, including: body mass index (BMI), as measured by a trained nurse; total energy intake (in Kcal); physical activity evaluated using the Physical Activity Scale for the Elderly (PASE) [17]; ethnicity; smoking habits, educational level and yearly income (< or ≥$50,000 or missing data); self-reported comorbidities that were assessed using the modified Charlson comorbidity score [18]; and, adherence to Mediterranean diet according to the score that was suggested by Panagiotakos et al. [19] and already reported in the OAI [14,20,21,22]. Data regarding diet, physical activity, demographics and comorbidities were recorded through questionnaires given to the participants. The covariates that were included in the multivariate analysis were those significantly different across dietary vitamin K quartiles (considering a *p*-value < 0.10) or significantly associated with depressive symptoms, according to a univariate analysis (*p*-value < 0.05).

### 2.5. Statistical Analyses

The normal distributions of continuous variables were tested using the Kolmogorov–Smirnov test. Data are shown as means ± standard deviations (SDs) for quantitative measures and the frequency and percentages for all discrete variables. For skewed distributions, the data were reported as median with the interquartile range, IQR. The P values for trends were calculated using the Jonckheere–Terpstra test for continuous variables and the Mantel–Haenszel Chi-square test for categorical variables. 

Several tests assessed the association between dietary vitamin K intake and depressive symptoms. First, we compared the mean values of CES-D across dietary vitamin K intake quartiles, using the p for trend for continuous variables. Second, we compared the dietary vitamin K intake between those with depressive symptoms vs. those without, while using an independent T-test. Finally, we used a logistic regression analysis using as exposure dietary vitamin K (categorized in quartiles or for each 100 ug/day increase) and as outcome the presence of depressive symptoms. The basic model was adjusted for age and sex. In addition to age and sex, the fully adjusted model, adjusted for: ethnicity (whites vs. others); BMI (as continuous); education (degree vs. others); smoking habits (current and previous vs. others); yearly income (categorized as ≥ or <50,000$, missing data); PASE (as continuous); Charlson co-morbidity index (as continuous); daily energy intake (as continuous); and, adherence to Mediterranean diet (as continuous). In all of the analyses, logistic regression data are reported as odds ratios (ORs) with 95% confidence intervals (CIs). Multi-collinearity among the covariates was assessed using the variance inflation factor (VIF), with a score of 2, leading to the exclusion of vitamin D supplementation, which had a high VIF with adherence to the Mediterranean diet (VIF = 2.13). In our analyses, we kept adherence to the Mediterranean diet as a covariate, since the strength of the association between depressive symptoms and adherence to Mediterranean diet was greater than that of vitamin D supplementation. We also explored in sensitivity analyses if gender (*p* = 0.08), BMI (categorized as overweight/obese vs. others) (*p* = 0.88), or vitamin D supplementation (test for interaction with a logistic regression analysis *p* < 0.001) can affect our results. Finally, a simple correlation analysis (Pearson’s coefficient) and a linear regression analysis were used and reported as R and standardized betas, respectively. 

All of the analyses were performed using SPSS 17.0 for Windows (SPSS Inc., Chicago, IL, USA). All of the statistical tests were two-tailed and statistical significance was assumed for a *p*-value < 0.05. 

## 3. Results

### 3.1. Sample Selection

The Osteoarthritis Initiative dataset initially included a total of 4,796 North American participants. At the baseline, we excluded 65 individuals, since they did not have data regarding depressive symptoms, 99 did not have data regarding their dietary vitamin K intake, and 257 with an implausible calorie intake (less than 800/greater than 4200 Kcal for men less than 500/greater than 3800 for women). Therefore, 4375 people were finally included in this research. 

In order to determine how representative our included sample was, we assessed for significant demographic and/or lifestyle differences between the included (*n* = 4375) and excluded (*n* = 421) participants. Excluded people did not differ in terms of mean age (*p* = 0.48), female gender prevalence (*p* = 0.29), depressive symptoms (in those missing vitamin K information, *p* = 0.84), or vitamin K intake (*p* = 0.28). 

### 3.2. Descriptive Characteristics 

Of the 4375 participants, 2538 were females and 1837 males. The mean age was 61.3 years (±9.2 years; range: 45–79). The median dietary vitamin K intake was 138 (IQR: 83–232) ug/day. 

Table 1 shows the participants’ characteristics by their dietary vitamin K intake. Those that were consuming more vitamin K (>232 ug/day) were more active, had a higher adherence to Mediterranean diet, consumed fruits, vegetables, and dairy products more frequently, and were more likely to be female, smokers, vitamin D supplementation consumers, and non-white than their counterparts (Table 1). On the contrary, no association between dietary vitamin K intake and age, education, income, BMI, or the presence of co-morbidity was found (Table 1). 

### 3.3. Dietary Vitamin K Intake and Depressive Symptoms

As shown in Table 1, subjects having a higher dietary vitamin K intake had significantly lower mean values of CESD (*p* for trend = 0.002). 

In the sample as whole, 437 (=10.0%) reported the presence of depressive symptoms. People with depressive symptoms reported significantly lower dietary vitamin K intake (165 ± 132 ug/day) when compared to those not having depressive symptoms (*n* = 3938) (185 ± 164 ug/day) (*p* = 0.001) (Figure 1). 

As shown in Table 2, the prevalence of depressive symptoms was significantly lower in people with higher dietary vitamin K intake (9.1% in those with a dietary vitamin K > 232 ug/day vs. 11.9% in those with <83 ug/day; *p* = 0.03). In the logistic regression analysis, after adjusting for 11 potential confounders and while taking people with lower dietary vitamin K intake as reference, people with the highest dietary vitamin K intake had significant lower odds of having depressive symptoms of about 42% (OR = 0.58; 95%CI: 0.43–0.80; *p* = 0.001) (Table 2). Each increase in 100 ug of dietary vitamin K intake was associated with significant lower odds of 12% (OR = 0.88; 95%CI: 0.82–0.95; *p* = 0.001) (*p* for trend = 0.003). However, this effect was only present in people not taking vitamin D as supplementation. In the people not taking vitamin D supplementation (*n* = 3166), in fact, we observed a significant lower odds of depressive symptoms in those having a higher vitamin K intake (Q4) (OR = 0.61; 95%CI: 0.43–0.88; *p* = 0.008), whilst the association between vitamin K intake and depression in those taking vitamin D supplementation (*n* = 1209) was not significant (Q4: OR = 0.72; 95%CI: 0.38–1.34; *p* = 0.28). Each increase in 100 ug of dietary vitamin K intake was only associated with a significant lower odds of 18% (OR = 0.82; 95%CI: 0.78–0.86; *p* < 0.001) (*p* for trend < 0.001) in people not taking vitamin D supplementation. 

Finally, we observed a weak negative correlation between dietary vitamin K intake and depressive symptoms (r = −0.15; *p* = 0.03), as confirmed by a linear regression analysis (standardized beta = −0.10; 95%CI: −0.17 to −0.03; *p* = 0.02). 

## 4. Discussion

In the present cross-sectional study, which included a large group of older adults from North America, we reported that depressive symptoms were significantly lower in people with higher dietary vitamin K intake. People in the highest quartile of vitamin K intake (i.e., >232 ug/day) had significantly lower odds of having depressive symptoms at baseline and each per 100 ug/day increment was associated with a significant lower odds of this condition of 12%. These findings remained unaltered after adjustment for potential confounders.

Persons with higher vitamin K intake were more frequently women and they had a higher physical activity levels, as evaluated with the PASE score. No differences were found regarding other potential risk factors for depression, such as age, educational level, income, and numbers of comorbidity. The finding that, in women, dietary intake of vitamin K was higher than in men is not surprising, since it is widely known that women more frequently consume vegetables and fruits than men (in our study, for example, women introduced a daily mean of 7.92 ± 4.64 g of vegetable fibers vs. 7.44 ± 4.56, *p* = 0.001), the major sources of vitamin K [23]. As higher levels of physical activity are closely related to muscle mass and strength, a better pattern of nutrition probably explains the higher level of physical activity in the group with higher vitamin K intake [24]. On the other hand, physical activity itself is an important factor in the prevention of development of depression, as suggested by a large literature regarding these topics [25,26,27,28,29]. However, we adjusted our analyses for the PASE score, suggesting that the association between dietary vitamin K and depressive symptoms could be independent from this factor. 

To the best of our knowledge, only one other study evaluated the relationship between the dietary intake of vitamin K and depression in older adults [5]. According to our findings, Nguyen et al. have found that, between several vitamins, only a lower vitamin K intake was associated with the presence of depressive symptoms, both in old men and women [5]. Conversely, no such association was found in children. This could be related to the difference in the kind of subjects that were enrolled in this study [11] and because the dietary intake of vitamin K in children is probably too low for reaching a biological effect. Vitamin K consumption, in fact, is usually doubled in adults than in children [5,11]. Moreover, the sample size that is included is limited to only 710 subjects, which probably leads to a type II error in their analyses. 

Moreover, our findings also resulted in agreement with studies on animals. Rats with vitamin K-deficiency presented hypoactivity, general malaise, and a lack of exploratory behavior [9]. There is also an indirect confirmation of our results in humans. Turker et al., for example, found that, in patients under treatment for atrial fibrillation with warfarin, which is a well-known vitamin K antagonist, the frequency of depression was significantly higher in period of warfarin use than after dabigatran transition [30].

From a pathophysiological point of view, lifetime low-vitamin K diet in rats is associated with higher levels of ceramides in the hippocampus [31]. High concentrations of ceramides have been related to pro-inflammatory processes, the production of reactive oxygen species, and the inhibition of neuronal survival [31]. A lack of neurogenesis in the hippocampus has been postulated as one of the possible pathogenetic causes of major depression. Consistent with this hypothesis, it was demonstrated that some of the antidepressants induce neurogenesis in the hippocampus and inhibit the ceramide system [32,33,34]. A meta-analysis on over 1300 subjects found that oxidative stress is increased in depression [35]. With such premises, we can speculate that a vitamin K supplementation might lower oxidative stress in the brain and therefore mitigate depressive symptoms.

Our findings have important clinical implications, which extend beyond increasing vitamin K intake for only depressive symptoms. This is because the consumption of vegetables, particularly green leafy ones, has been related to a protective effect against cardiovascular disease [36], lower oxidative stress improving antioxidant status [37], and reduced visceral and liver fat and risk factors for type 2 diabetes [38]. Major food sources of vitamin K include vegetables, especially green leafy vegetables, vegetable oils, and some fruits. The most common sources of vitamin K in the United States (U.S.) diet are spinach, broccoli, iceberg lettuce, and fats and oils, particularly soybean and canola oil [39].

Finally, another important finding of our research is that vitamin K was associated with depressive symptoms, only in people not taking vitamin D supplementation. While a recent paper suggest that vitamin D supplementation might lower vitamin K status [40], in our study, vitamin D supplementation seems to attenuate the negative effect of a low vitamin K intake. Nevertheless, in our study, another limitation is the lack of measurement of the uncarboxylated fraction of osteocalcin and matrix gla protein, markers of vitamin K status, which could better account for the real effect of vitamin K on depression and bone health. These data indicated that further research is needed to better elucidate the important issue of the interaction between vitamin D and K that seem to have important antidepressant effects. 

The findings of our work should be interpreted within its limitations. First, the cross-sectional nature of the work can introduce a reverse causality probability, i.e., people having depression consume less frequently foods rich of vitamin K, such as vegetables. [41] Second, the OAI only includes people having knee OA or that are at a high risk of this condition. Thus, further research is needed to see if our findings can be generalized to the general population. People having knee OA are at higher risk of depression than the general population, and therefore it is possible that we found stronger results, since the prevalence of depressive symptomatology is higher than expected [16,42]. However, in which direction this bias can affect our results is hard to say. Third, vitamin K intake was estimated by a retrospective self-report of eating habits over one year, and it thus potentially carries itself a risk of imprecision, inaccuracy, and bias (e.g., social desirability bias). Fourth, information regarding medications (and in particularly vitamin K antagonists and antidepressants) that might interfere in the association that we found between vitamin K and depressive symptoms were not included. Fifth, while there is some debate about whether the effects of different types of vitamin K (i.e., K1 or K2) have different effects on human health, the data available for this study only allowed for us to explore relationships with vitamin K as a whole with depressive symptoms [43]. Finally, depressive mood was only assessed with the CESD scale, whilst the gold standard for the diagnosis of depression is a structured interview with an expert of this disease.

## 5. Conclusions

Higher dietary vitamin K intake was significantly associated with a lower presence of depressive symptoms, also after accounting for potential confounders, suggesting a role for this vitamin in the prevention and treatment of depressed mood. However, future longitudinal and intervention studies are needed to confirm or refute our findings.

## Figures and Tables

**Figure 1 nutrients-11-00787-f001:**
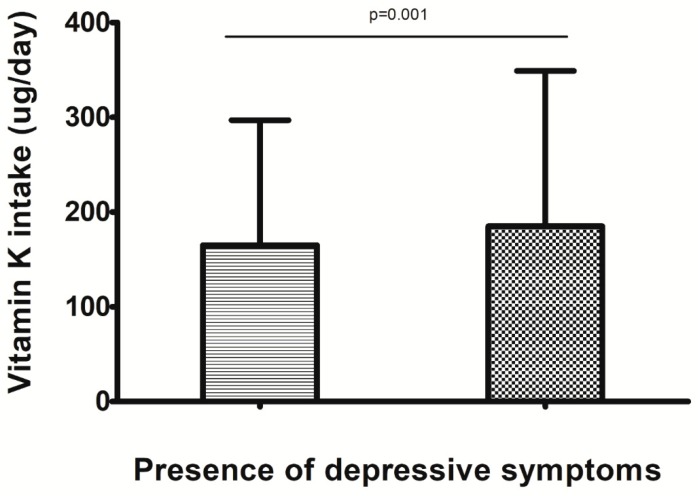
The data are reported as means with their standard deviations. Left column = individuals with depressive symptoms; right column = individuals without depressive symptoms.

**Table 1 nutrients-11-00787-t001:** Characteristics of the participants classified according to their dietary vitamin K intake.

	Vitamin K <83 µg (*n* = 1094)	Vitamin K 83–138 µg (*n* = 1094)	Vitamin K 139–232 µg (*n* = 1094)	Vitamin K > 232 µg (*n* = 1093)	*p* Value for Trend ^1^
Energy intake (Kcal/day)	1222 (455)	1377 (487)	1479 (536)	1642 (621)	<0.001
Fiber (g/day)	4.1 (2.2)	6.1 (2.4)	8.1 (2.8)	12.5 (5.2)	<0.001
Fruits (servings/day)	1.12 (0.77)	1.43 (0.83)	1.58 (0.89)	1.72 (0.97)	<0.001
Vegetables (servings/day)	1.47 (0.77)	2.48 (0.88)	3.62 (1.20)	6.05 (2.61)	<0.001
Dairy (servings/day)	1.25 (0.95)	1.35 (0.93)	1.42 (0.97)	1.44 (1.03)	<0.001
aMED (points)	26 (5)	28 (6)	29 (5)	30 (5)	<0.001
Age (years)	60.1 (9.5)	61.5 (9.1)	61.6 (9.4)	61.0 (8.8)	0.21
PASE (points)	156 (81)	158 (79)	161 (82)	167 (84)	0.007
Females (*n*, %)	546 (49.9)	595 (54.4)	657 (60.1)	740 (67.7)	<0.001
White race (*n*, %)	898 (82.2)	929 (84.9)	898 (82.2)	790 (72.3)	<0.001
Smoking (previous/current) (*n*, %)	489 (45.0)	488 (44.9)	527 (48.3)	560 (51.5)	0.001
Graduate degree (*n*, %)	291 (26.6)	363 (33.2)	332 (30.4)	341 (31.2)	0.07
Yearly income (≥$50,000)	613 (56.0)	650 (59.4)	686 (62.7)	635 (58.1)	0.13
Vitamin D supplementation (*n*, %)	228 (21.2)	292 (27.1)	332 (30.9)	357 (33.0)	<0.001
BMI (Kg/m^2^)	28.5 (4.7)	28.7 (4.8)	28.6 (4.8)	28.9 (4.9)	0.34
Charlson co-morbidity index (points)	0.41 (0.91)	0.40 (0.87)	0.37 (0.79)	0.39 (0.79)	0.58
CESD points (SD)	7.3 (7.8)	6.3 (6.7)	6.4 (6.5)	6.4 (6.8)	0.002

Notes: The data are presented as means (with standard deviations) for continuous variables and number (with percentage). ^1^
*p* values for trends were calculated using the Jonckheere-Terpstra test for continuous variables and the Mantel-Haenszel Chi-square test for categorical variables. Abbreviations: aMED: adherence to Mediterranean diet; BMI: body mass index; CESD: Center for Epidemiologic Studies-Depression; PASE: Physical Activity Scale for the Elderly.

**Table 2 nutrients-11-00787-t002:** Association between dietary vitamin K intake and depressive symptoms.

	Whole Sample (*n* = 4375)					Not Taking Vitamin D Supplementation (*n* = 3166)				Taking Vitamin D Supplementation (*n* = 1209)			
	Prevalence Depressive Symptoms (%)	Basic-Adjusted ^1^ OR (95%CI)	*p* Value	Fully-Adjusted ^2^ OR (95%CI)	*p* Value	Basic-Adjusted ^1^ OR (95%CI)	*p* Value	Fully-Adjusted ^2^ OR (95%CI)	*p* Value	Basic-Adjusted OR (95%CI)	*p* Value	Fully-Adjusted ^2^ OR (95%CI)	*p* Value
Vitamin K <83 µg	130/1094 (=11.9)	1 [reference] [*p* for trend = 0.04]		1 [reference] [*p* for trend = 0.02]		1 [reference] [*p* for trend = 0.03]		1 [reference] [*p* for trend = 0.02]		1 [reference] [*p* for trend = 0.07]		1 [reference] [*p* for trend = 0.08]	
Vitamin K 83–138 µg	105/1094 (=9.6)	0.79 (0.60–1.04)	0.10	0.80 (0.60–1.07)	0.13	0.66 (0.47–0.92)	0.01	0.65 (0.46–0.93)	0.02	1.24 (0.70–2.20)	0.45	1.15 (0.63–2.10)	0.65
Vitamin K 139–232 µg	102/1094 (=9.3)	0.74 (0.56–0.98)	0.04	0.76 (0.57–1.03)	0.08	0.80 (0.58–1.11)	0.18	0.85 (0.59–1.21)	0.36	0.64 (0.34–1.20)	0.17	0.52 (0.26–1.03)	0.06
Vitamin K >232 µg	100/1093 (=9.1)	0.67 (0.51–0.89)	0.006	0.58 (0.42–0.81)	0.001	0.64 (0.45–0.90)	0.01	0.61 (0.43–0.88)	0.008	0.69 (0.38–1.25)	0.22	0.72 (0.38–1.34)	0.28

Notes: ^1^ Basic-adjusted model included as covariates age (as continuous) and sex. ^2^ Fully-adjusted model included as covariates: age (as continuous); sex; race (whites vs. others); body mass index (as continuous); education (degree vs. others); smoking habits (current and previous vs. others); yearly income (categorized as ≥ or <50,000$ and missing data); Physical Activity Scale for Elderly score (as continuous); Charlson co-morbidity index (as continuous); daily energy intake (as continuous); adherence to Mediterranean diet (as continuous). Abbreviations: CI: confidence intervals; OR: odds ratio.

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
