# Peer review of "The Relationship between Dietary Vitamin K and Depressive Symptoms in Late Adulthood: A Cross-Sectional Analysis from a Large Cohort Study"

_nutrients, 2019, doi:10.3390/nu11040787_

Round 1
Reviewer 1 Report
Line 101 - There is no information on how the intake of vitamin K in the diet was calculated.
Author Response
Reviewer 1.
Line 101 - There is no information on how the intake of vitamin K in the diet was calculated.
Reply: We have added this sentence for better explaining how vitamin K was calculated (lines 100-107):
“The vitamin K intake was then calculated as sum of vitamin K supplementation (if any) and of dietary intake of vitamin K as assessed by the diet recall questionnaire, the Block Brief 2000 FFQ. The vitamin K scores for this study were was calculated automatically from the Block Brief 2000 FFQ by NutritionQuest (http://www.nutritionquest.com/assessment/), using standard values for amounts of vitamin K across an array of foods. The foods with particularly high reference levels of vitamin K are leafy vegetables, such as kale, broccoli and spinach, with lower (but still notable) levels in fruits and berries and fish.”
Reviewer 2 Report
This is a cross-sectional analysis that investigated vitamin K intake in relation to depressive symptoms in a cohort of 4,375 participants from the Osteoarthritis Initiative. This topic is interesting, but much more information about vitamin K intake should be provided. (main foods, vitamin K1/vitamin K2).
Abstract:
Don’t use the word “Decreased” for a cross-sectional study, but use the words ‘was lower’
Rewrite: People with depressive symptoms had a significant lower dietary vitamin K intake than their counterparts (165±132 vs. 185±164 ug/day p=0.001). It looks that the direction is depressive symptoms --> lower vitamin K.
Methods
Exposure – vitamin K
In the description of food intake to vitamin K the calculation method is missing. For instance a food composition table and the source. Is the FFQ validated to measure vitamin K intake? Was vitamin K assessed as vitamin K1 and vitamin K2? Please, elaborate.
Increase in 100ug/day should be per 100 ug/day increment
Provide more food/nutrient data. What about fiber intake to check vitamin K1 intake, what about dairy to check vitamin K2 intake. Provide more detailed nutrient and food data.
The covariates should be described in more detail. How have they been measured?
Results
Were the 4,796 participants different from the 4375 included participants. Please include the main characteristics: age, sex, etc.
Vitamin K intake was higher among women. Please provide food data as well to check if vegetables intake was indeed higher.
What if the results between vitamin K were analyzed continuously for depressive symptoms?
The authors adjusted the analyses for a lot of confounders, but they have not been listed in table 1. Please, provide this information for a better understanding of the population.
What about medication use. Bisphosphonates, vitamin D? Included this important information.
Include results for vitamin K1 and vitamin K2 with depression as separate analyses.
Tables: the N per category of vitamin K should be provided.
Figure 1: please provide the mean values and range in a footnote.
Discussion
Please, elaborate on the potential mechanism. How would vitamin K lower depressive symptoms? Include this in the discussion.
Recent evidence suggest that vitamin D supplementation might lower vitamin K status. Nutrients 2019, 11(2), 231; https://doi.org/10.3390/nu11020231.Please, elaborate on the potential mechanism and mention as limitation that a marker of vitamin K status has not been taken into account (matrix gla protein).
Author Response
Reviewer 2
This is a cross-sectional analysis that investigated vitamin K intake in relation to depressive symptoms in a cohort of 4,375 participants from the Osteoarthritis Initiative. This topic is interesting, but much more information about vitamin K intake should be provided. (main foods, vitamin K1/vitamin K2).
Reply: Good point. We have added how vitamin K was calculated according to the indications of the OAI guidelines (lines 100-107):
“The vitamin K intake was then calculated as sum of vitamin K supplementation (if any) and of dietary intake of vitamin K as assessed by the diet recall questionnaire, the Block Brief 2000 FFQ. The vitamin K scores for this study were was calculated automatically from the Block Brief 2000 FFQ by NutritionQuest (http://www.nutritionquest.com/assessment/), using standard values for amounts of vitamin K across an array of foods. The foods with particularly high reference levels of vitamin K are leafy vegetables, such as kale, broccoli and spinach, with lower (but still notable) levels in fruits and berries and fish. ”
Abstract:
Don’t use the word “Decreased” for a cross-sectional study, but use the words ‘was lower’.
Reply: Done in the Abstract and in the manuscript.
Rewrite: People with depressive symptoms had a significant lower dietary vitamin K intake than their counterparts (165±132 vs. 185±164 ug/day p=0.001). It looks that the direction is depressive symptoms --> lower vitamin K.
Reply: We removed this sentence since could be misinterpreted.
Methods
Exposure – vitamin K
In the description of food intake to vitamin K the calculation method is missing. For instance a food composition table and the source. Is the FFQ validated to measure vitamin K intake? Was vitamin K assessed as vitamin K1 and vitamin K2? Please, elaborate.
Reply: Vitamin K intake was not directly calculated by us, but already reported in the OAI databases. Despite several attempts in asking them this specific question, we did not obtain any answer re vitamin K1 and K2. We have reported in the Methods section how vitamin K was calculated, according to the information that we have (Methods section, lines 100-107):
“The vitamin K intake was then calculated as sum of vitamin K supplementation (if any) and of dietary intake of vitamin K as assessed by the diet recall questionnaire, the Block Brief 2000 FFQ. The vitamin K scores for this study were was calculated automatically from the Block Brief 2000 FFQ by NutritionQuest (http://www.nutritionquest.com/assessment/), using standard values for amounts of vitamin K across an array of foods. The foods with particularly high reference levels of vitamin K are leafy vegetables, such as kale, broccoli and spinach, with lower (but still notable) levels in fruits and berries and fish.”
Increase in 100ug/day should be per 100 ug/day increment
Reply: Done.
Provide more food/nutrient data. What about fiber intake to check vitamin K1 intake, what about dairy to check vitamin K2 intake. Provide more detailed nutrient and food data.
Reply: Good point. However, we would like to observe that both fiber and dairy intake are already included in the MD calculation. Therefore, to adjust also for these foods can result in an over-adjustment of our analyses.
The covariates should be described in more detail. How have they been measured?
Reply: We have modified this paragraph as follows, also adding that BMI was measured by a trained nurse.
“Data regarding diet, physical activity, demographics and comorbidities were recorded through questionnaires given to the participants.”
Results
Were the 4,796 participants different from the 4375 included participants. Please include the main characteristics: age, sex, etc.
Reply: Good point. We have added at line 160-164 this consideration:
“In order to determine how representative our included sample was, we assessed for significant demographic and/or lifestyle differences between included (n=4375) and excluded (n=421) participants. Excluded people did not differ in terms of mean age (p=0.48), female gender prevalence (p=0.29), depressive symptoms (in those missing vitamin K information, p=0.84) or vitamin K intake (p=0.28).”
Vitamin K intake was higher among women. Please provide food data as well to check if vegetables intake was indeed higher.
Reply: We have added a sentence in the Discussion, regarding this specific point, as follows (lines 227-230):
“The finding that in women dietary intake of vitamin K was higher than in men is not surprising, since it is widely known that women consume more frequently vegetables and fruits than men (in our study, for example, women introduced a daily mean of 7.92±4.64 g of vegetable fibers vs. 7.44±4.56, p=0.001), the major sources of vitamin K.”
What if the results between vitamin K were analyzed continuously for depressive symptoms?
Reply: We have added a sentence regarding this analysis in the Results section, as follows:
“Finally, in a post-hoc analyses examining the relationships between vitamin K intake and depressive symptoms on a continuous scale, we observed a significant negative correlation between the two, with higher levels of vitamin K intake associated with fewer depressive symptoms (r=-0.15; p=0.03).”
The authors adjusted the analyses for a lot of confounders, but they have not been listed in table 1. Please, provide this information for a better understanding of the population.
Reply: Thank you so much for your careful reading. We have added the adherence to Mediterranean diet as missing in the first submission.
What about medication use. Bisphosphonates, vitamin D? Included this important information.
Reply: We kindly disagree with the Reviewer. Even if this information is available in the OAI, we believe that these confounders are of poor relevance in the association between vitamin K and depression, whilst they could be very important in the association between vitamin K and bone health outcomes.
Include results for vitamin K1 and vitamin K2 with depression as separate analyses.
Reply: Unfortunately, we don’t have this information and, consequently, we have added this as possible limitation of our paper in the Discussion section as follows (lines 285-288):
“Fifth, while there is some debate about whether the effects of different types of vitamin K (i.e. K1 or k2) have different effects on human health[40], the data available for this study only allowed us to explore relationships with vitamin K as a whole with depressive symptoms.”
Tables: the N per category of vitamin K should be provided.
Reply: In both tables the N of each vitamin K category it is reported.
Figure 1: please provide the mean values and range in a footnote.
Reply: The mean values are reported in the text, lines 169-170.
Discussion
Please, elaborate on the potential mechanism. How would vitamin K lower depressive symptoms? Include this in the discussion.
Reply: We sincerely thank the Reviewer for this comment. In the previous version, we have already discussed this point as follows:
“From a pathophysiological point of view, preclinical studies have demonstrated that lifetime low-vitamin K diet are associated with higher levels of ceramides in the hippocampus. [32] High concentrations of ceramides have been related to pro-inflammatory processes, production of reactive oxygen species and inhibition of neuronal survival. [32] A lack of neurogenesis in the hippocampus has been postulated as one of the possible pathogenetic causes of major depression. Consistent with this hypothesis, it was demonstrated that some antidepressants induce neurogenesis in the hippocampus and inhibit the ceramide system. [33-35] “
In order to answer to your question, we have added the following sentences (lines 257-259):
“A meta-analysis on over 1300 subjects found that oxidative stress is increased in depression.[36] With such premises we can speculate that a vitamin K supplementation might lower oxidative stress in brain and therefore mitigate depressive symptoms.”
Recent evidence suggest that vitamin D supplementation might lower vitamin K status. Nutrients 2019, 11(2), 231; https://doi.org/10.3390/nu11020231.Please, elaborate on the potential mechanism and mention as limitation that a marker of vitamin K status has not been taken into account (matrix gla protein).
Reply: Good point. We have tested the interaction vitamin D supplementation by vitamin K intake in being associated with depressive symptoms in our study. As you stated, we observed a significant interaction between vitamin D supplementation and vitamin K intake, suggesting that the effect of vitamin K on depression is limited only in case of the absence of vitamin D supplementation.We have added these relevant Results as follows (lines 186-193):“This effect, however, was present only in people not taking vitamin D as supplementation. In the people not taking vitamin D supplementation, in fact, we observed a significant lower odds of depressive symptoms in those having a higher vitamin K intake (Q4) (OR=0.61; 95%CI: 0.43-0.88; p=0.008), whilst the association between vitamin K intake and depression in those taking vitamin D supplementation was not significant (Q4: OR=0.72; 95%CI: 0.38-1.34; p=0.28). Each increase in 100 ug of dietary vitamin K intake was associated with a significant lower odds of 18% (OR=0.82; 95%CI: 0.78-0.86; p<0.001) (p for trend<0.001), only in people not taking vitamin D supplementation.”
We have consequently added a comment in the Discussion section (lines 267-273), as follows:
“Finally, another important finding of our research is that vitamin K was associated with depressive symptoms, only in people not taking vitamin D supplementation. While a recent paper suggest that vitamin D supplementation might lower vitamin K status [41], in our study vitamin D supplementation seem to override the negative effect of a low vitamin K intake. Nevertheless, in our study another limitation is the lack of measurement of the uncarboxylated fraction of osteocalcin and matrix gla protein, markers of vitamin K status, that could better account for the real effect of vitamin K on depression and bone health.”
Reviewer 3 Report
The study of Bolzetta et al. investigates the association between dietary vitamin K and depression using a through a semi-quantitative food frequency questionnaire in a large cohort of 4,375 participants aged 45-79 years from the Osteoarthritis Initiative. Depressive symptoms were diagnosed using the 20-item Center for Epidemiologic Studies-Depression (CES-D) > 16.
As the authors mention in the last part of the discussion, there are serious limitations in the study, as it relays on a questionnaire for vitamin K status, without any actual measurements. Therefore, the study cannot differentiate between vitamin K and other benefits from vitamin K-containing nutrients. Further, the cohort does not seem to include medication of this patients. Given the age of the group, it is possible that an important fraction is taking oral anticoagulants, which would interfere with vitamin K metabolism. This is an important weakness in the study design.
As mention by the authors, a limitation of the study is that the cohort consists of people having knee osteoarthritis or at high risk of this condition. Could the authors speculate on how this could bias the results in their opinion?
Considering the results of the study in reference 5, it would be interesting to know the results obtained separately in men and women and in the group of overweight people. This factors are taking into account on the multivariate analysis, but still could be of interest to know if the effect of vitamin K intake as measured in the study is different in these groups.
There are some typos or unclear expressions, including:
Line 67 “may not provide”?
Line 72 “a dietary model”
Line 75 “On the contrary”
Lines 187 to 189. Include information of each group in the X axis. The legend of the figure seems misplaced. The discussion section should start in a new paragraph.
Line 200 “higher attitude”?
Line 223 A reference is missing on the relation of ceramide concentration and vitamin K intake.
Line 231 “related to a protective”
Author Response
Reviewer 3
As the authors mention in the last part of the discussion, there are serious limitations in the study, as it relays on a questionnaire for vitamin K status, without any actual measurements. Therefore, the study cannot differentiate between vitamin K and other benefits from vitamin K-containing nutrients. Further, the cohort does not seem to include medication of this patients. Given the age of the group, it is possible that an important fraction is taking oral anticoagulants, which would interfere with vitamin K metabolism. This is an important weakness in the study design.
Reply: We thank the Reviewer for these considerations. As she/he already said, we have already acknowledged these as limitations of our paper. Moreover, as the other reviewers asked, we have added other limitations, e.g. the impossibility to explore the effect of vitamin K1 and K2 on depression.
As mention by the authors, a limitation of the study is that the cohort consists of people having knee osteoarthritis or at high risk of this condition. Could the authors speculate on how this could bias the results in their opinion?
Reply: We have added this sentence in the Discussion section (lines 278-281):
“People having knee OA are at higher risk of depression than general population and therefore it is possible that we found stronger results since the prevalence of depressive symptomatology is higher than expected. [16,41]”
Considering the results of the study in reference 5, it would be interesting to know the results obtained separately in men and women and in the group of overweight people. This factors are taking into account on the multivariate analysis, but still could be of interest to know if the effect of vitamin K intake as measured in the study is different in these groups.
Reply: Good point. In order to satisfy your request, we have tested the interaction between these factors (sex and BMI) by quartiles of vitamin K without finding any significant association. We have now reported in the Statistical analysis section this concept, as follows (lines 147-150):
“We also explored in sensitivity analyses if gender (p=0.08) and BMI (categorized as overweight/obese vs. others) (p=0.88) can affect our results.”
There are some typos or unclear expressions, including:
Line 67 “may not provide”?
Line 72 “a dietary model”
Line 75 “On the contrary”
Reply: Corrected.
Lines 187 to 189. Include information of each group in the X axis. The legend of the figure seems misplaced. The discussion section should start in a new paragraph.
Reply: Done.
Line 200 “higher attitude”?
Reply: We have changed this sentence as follows:
“The finding that in women dietary intake of vitamin K was higher than in men is not surprising, since it is widely known that women consume more frequently vegetables and fruits than men…”
Line 223 A reference is missing on the relation of ceramide concentration and vitamin K intake.
Reply: We have added a new reference for explaining the possible association between vitamin K and ceramides, as requested.
Line 231 “related to a protective”
Reply: Done.
Round 2
Reviewer 2 Report
The relationship between dietary vitamin K and 2 depressive symptoms in late adulthood: a cross-sectional analysis from a large cohort study
The authors updated the manuscript and the paper improved, however crucial information about the study population is missing and should be added. Results need to be interpreted and some things should be explored in more detail. Further, the manuscript needs some grammar and spelling checks throughout the paper.
The vitamin K calculations have been added. Please, indicate how many recall days were used and if this is a good method to estimate vitamin K intake. Was this method validated?
Provide more food/nutrient data. What about fiber intake to check vitamin K1 intake, what about dairy to check vitamin K2 intake. Provide more detailed nutrient and food data in table 1. The question is not to adjust for these factors, but to provide the reader with more information about foods related to vitamin K intake such as fruits and vegetables, fiber, dairy intake. Please, add these covariates to table 1. Based on this information one can decide to adjust for such a factor.
What if the results between vitamin K were analyzed continuously for depressive symptoms?
Please, provide regression analyses and not univariate correlation coefficients and add this to table 2.
What about medication use. Bisphosphonates, vitamin D? This information is very valuable to get a better understanding of the study population. Please, add this information. Vitamin D is related to depressive symptoms so it is very valuable to check whether vitamin D is a confounder. The readers need to get a better understanding of the population.
There appears to be significant effect modification by vitamin D supplementation. What was the P-for interaction and how was it tested for? It is crucial to mention this in table 1 and provide separate tables for no vitamin D use and vitamin D use. This changes the entire conclusion of the paper and should not be hidden in the text. It is still unclear how many participants used vitamin D supplements?
In the people not taking vitamin D supplementation, in fact, we observed a significant lower odds of depressive symptoms in those having a higher vitamin K intake (Q4) (OR=0.61; 95%CI: 0.43-0.88; p=0.008), whilst the association between vitamin K intake and depression in those taking vitamin D supplementation was not significant (Q4: OR=0.72; 95%CI: 0.38-1.34; p=0.28). Each increase in 100 ug of dietary vitamin K intake was associated with a significant lower odds of 18% (OR=0.82; 95%CI: 0.78-0.86; p<0.001) (p for trend<0.001), only in people not taking vitamin D supplementation.”
Please add all results per quartile, (not only highest quartile). This deserves a separate table and should be interpreted. Why is this the case? Is this expected? What new research questions can be generated based on this result?
The conclusion “in our study vitamin D supplementation seem to override the negative effect of a low vitamin K intake” should be toned down. Based on yes/no supplement use, this cannot be stated. Please, elaborate on the potential mechanisms. Were the people on vitamin D supplements more frail/sicker? These questions should be incorporated.
Author Response
Reviewer 2
The authors updated the manuscript and the paper improved, however crucial information about the study population is missing and should be added. Results need to be interpreted and some things should be explored in more detail. Further, the manuscript needs some grammar and spelling checks throughout the paper.
R: The paper was revised by an English native speaker and the Results were better approached also through your comments.
The vitamin K calculations have been added. Please, indicate how many recall days were used and if this is a good method to estimate vitamin K intake. Was this method validated?
R: this is a food frequency questionnaire and not a 24 hr or 48 hr recall. We have changed this part (lines 97-100), as follows:
“The semi-quantitative Block Brief 2000 food frequency questionnaire (FFQ) at baseline, a validated and widely used method [13,14] was used for collecting information regarding dietary habits over the previous year.”
Provide more food/nutrient data. What about fiber intake to check vitamin K1 intake, what about dairy to check vitamin K2 intake. Provide more detailed nutrient and food data in table 1. The question is not to adjust for these factors, but to provide the reader with more information about foods related to vitamin K intake such as fruits and vegetables, fiber, dairy intake. Please, add these covariates to table 1. Based on this information one can decide to adjust for such a factor.
R: as requested we have added this information for descriptive results. Regarding if to adjust or not we would like to report that the analyses are adjusted for the adherence to Mediterranean diet that includes all these factors.
What if the results between vitamin K were analyzed continuously for depressive symptoms?
Please, provide regression analyses and not univariate correlation coefficients and add this to table 2.
R: as requested we have added this information in the text as follows (lines 197-200):
“Finally, we observed a weak negative correlation between dietary vitamin K intake and depressive symptoms (r=-0.15; p=0.03), confirmed by a linear regression analysis (standardized beta=-0.10; 95%CI: -0.17 to -0.03; p=0.02).”
What about medication use. Bisphosphonates, vitamin D? This information is very valuable to get a better understanding of the study population. Please, add this information. Vitamin D is related to depressive symptoms so it is very valuable to check whether vitamin D is a confounder. The readers need to get a better understanding of the population.
R: good point. We have now reported the prevalence of vitamin D supplementation consumers in Table 1 and the number of people taking or not vitamin D in the text, as requested. Vitamin D supplementation, however, was removed from the covariates for which we adjusted our analyses, as follows (lines 146-150):
“Multi-collinearity among covariates was assessed using the variance inflation factor (VIF), with a score of 2, leading to the exclusion of vitamin D supplementation, that had a high VIF with adherence to Mediterranean diet adherence (VIF=2.13). In our analyses, we kept adherence to Mediterranean diet as covariate, since the strength of the association between depressive symptoms and adherence to Mediterranean diet was greater than that of vitamin D supplementation.”
There appears to be significant effect modification by vitamin D supplementation. What was the P-for interaction and how was it tested for? It is crucial to mention this in table 1 and provide separate tables for no vitamin D use and vitamin D use. This changes the entire conclusion of the paper and should not be hidden in the text. It is still unclear how many participants used vitamin D supplements?
R: good point. We have added this information in the text (in terms of number of people taking or not vitamin D supplementation) adding also the information required for the p for interaction. Regarding the division of Table 1 in those or not taking vitamin D supplementation we kindly disagree with the Reviewer’s opinion. We have now entirely reported the information divided by vitamin D supplementation, regarding the association between vitamin K and depressive symptoms, as you requested below. To add descriptive findings is in our opinion, not useful for the reader, but we are keen to know what the EIC says regarding this specific point.
In the people not taking vitamin D supplementation, in fact, we observed a significant lower odds of depressive symptoms in those having a higher vitamin K intake (Q4) (OR=0.61; 95%CI: 0.43-0.88; p=0.008), whilst the association between vitamin K intake and depression in those taking vitamin D supplementation was not significant (Q4: OR=0.72; 95%CI: 0.38-1.34; p=0.28). Each increase in 100 ug of dietary vitamin K intake was associated with a significant lower odds of 18% (OR=0.82; 95%CI: 0.78-0.86; p<0.001) (p for trend<0.001), only in people not taking vitamin D supplementation.”
Please add all results per quartile, (not only highest quartile). This deserves a separate table and should be interpreted. Why is this the case? Is this expected? What new research questions can be generated based on this result?
R: we have added these elaborations in Table 2, as you suggested. Regarding the interpretation of these results, we have tried to explain in lines 268-275 and you have now requested, we have added a sentence regrading the research questions as follows (lines 275-278):
“These data indicated that further research is needed to better approach the important issue of the interaction between vitamin D and K that seem to have important antidepressant effects.”
The conclusion “in our study vitamin D supplementation seem to override the negative effect of a low vitamin K intake” should be toned down. Based on yes/no supplement use, this cannot be stated. Please, elaborate on the potential mechanisms. Were the people on vitamin D supplements more frail/sicker? These questions should be incorporated.
R: we have changed the word override with the word attenuate, in order to satisfy your request. The potential mechanisms regarding vitamin K and D are, in our opinion, explained, with a particular focus on animal models. Finally, regarding your request on vitamin D supplementation groups’ difference, we failed to observe any significant difference in terms of Charlson comorbidity index between vitamin D users or not (p=0.48, Student t-test). Therefore, we did not incorporate these considerations in the paper.
Reviewer 3 Report
The authors have indicated more accurately the limitations of the study. They have analyzed further the data set. The study has improved.
Author Response
Thank you to the Reviewer for her/his valuable comments.